# A gradient oxy-thiophosphate-coated Ni-rich layered oxide cathode for stable all-solid-state Li-ion batteries

Jianwen Liang[1,6], Yuanmin Zhu[2,3,6], Xiaona Li[1,6], Jing Luo[1], Sixu Deng[1], Yang Zhao ●[1], Yipeng Sun[1], Duojie Wu[3], Yongfeng Hu[4], Weihan Li[1,5], Tsun-Kong Sham ●[5], Ruying Li[1], Meng Gu ●[3] ✉ & Xueliang Sun ●[1] ✉

High-energy Ni-rich layered oxide cathode materials such as $LiNi_{0.8}Mn_{0.1}Co_{0.1}O_2$ (NMC811) suffer from detrimental side reactions and interfacial structural instability when coupled with sulfide solid-state electrolytes in all-solid-state lithium-based batteries. To circumvent this issue, here we propose a gradient coating of the NMC811 particles with lithium oxy-thiophosphate ($Li_3P_{1+x}O_4S_{4x}$). Via atomic layer deposition of $Li_3PO_4$ and subsequent in situ formation of a gradient $Li_3P_{1+x}O_4S_{4x}$ coating, a precise and conformal covering for NMC811 particles is obtained. The tailored surface structure and chemistry of NMC811 hinder the structural degradation associated with the layered-to-spinel transformation in the grain boundaries and effectively stabilize the cathode|solid electrolyte interface during cycling. Indeed, when tested in combination with an indium metal negative electrode and a $Li_{10}GeP_2S_{12}$ solid electrolyte, the gradient oxy-thiophosphate-coated NCM811-based positive electrode enables the delivery of a specific discharge capacity of 128 mAh/g after almost 250 cycles at $0.178 \, mA/cm^2$ and 25 °C.

The development of highly stable energy storage systems is an essential subject to solve the current energy challenges. Although conventional non-aqueous liquid electrolyte-based lithium-ion batteries (LIBs) can serve as a power source for many modern applications, there have been gradually raised safety concerns due to the use of flammable organic liquid electrolytes. Solidifying LIBs by substituting the liquid organic electrolytes with solid-state electrolytes (SSEs) to fabricate all-solid-state lithium batteries (ASSLBs) is considered a promising approach due to the significantly improved safety and high theoretical energy density[1–5]. The promise of ASSLBs has stimulated extensive research for the development of ionic conductive SSEs and the successful implementation of high-voltage oxide cathode materials to meet the increasing demands of high-energy-density ASSLBs[6–12].

Among various types of SSEs, sulfide SSEs are promising due to their high ionic conductivity up to $10^{-2} \, S \, cm^{-1}$ at 25 °C, high cation transport number (å 0.9), and good mechanical deformability[13–17]. As appealing cathode materials, the layered oxide cathodes, especially the Ni-rich NMC cathodes (e.g. $LiNi_{0.8}Mn_{0.1}Co_{0.1}O_2$, NMC811), stand out to compete with the state-of-the-art LIBs in terms of high capacity and high energy density[18–21]. However, the integration of sulfide-based ASSLBs with Ni-rich oxide cathodes still encounters severe challenges: 1) decomposition of sulfide SSEs at high voltages because of their limited thermodynamic electrochemical stability window; 2) parasitic

[1]Department of Mechanical and Materials Engineering, University of Western Ontario, London, ON N6A 5B9, Canada. [2]Research Institute of Interdisciplinary Science & School of Material Science and Engineering, Dongguan university of technology, Dongguan, Guangzhou, China. [3]Department of Materials Science and Engineering, Southern University of Science and Technology, Shenzhen 518055, China. [4]Canadian Light Source, 44 Innovation Boulevard, Saskatoon, SK S7N 2V3, Canada. [5]Department of Chemistry, University of Western Ontario, London, ON N6A 5B9, Canada. [6]These authors contributed equally: Jianwen Liang, Yuanmin Zhu, Xiaona Li. ✉e-mail: gum@sustech.edu.cn; xsun9@uwo.ca

interfacial reactions between sulfide SSEs and NMC811 upon contact and formation of ionic insulating decomposition products; 3) formation of space-charge layer (SCL) between sulfide SSEs and oxide cathodes due to their unmatch chemical potentials, where the Li⁺ ions near the interface are redistributed resulting in a high-resistant Li depletion layer at the sulfide SSE side; 4) capacity and voltage decay issues from as the structural degradation occurs at the surface and grain boundaries of the Ni-rich oxide cathode particles[22–26]. All these issues should be addressed at the same time to achieve stable and reliable ASSLBs.

Constructing an artificial coating on the cathode particles is a promising approach. Various coating materials (such as $Al_2O_3$, $Li_2CO_3$, $LiNbO_3$, $LiNb_{0.5}Ta_{0.5}O_3$, $Li_3PO_4$, $Li_3BO_3$, and conductive polymer)[11,22,27–33] have been fabricated by atomic layer deposition (ALD)[29,31], pulsed laser deposition (PLD)[22], chemical vapour deposition (CVD)[30] or sol-gel[27,28] method. These attempts have been demonstrated to be effective in improving the electrochemical performance of ASSLBs. However, for high-performance and long-cycling ASSLBs, the artificial coating must possess multiple functions including protection of sulfide SSEs from decomposition, stabilization of the cathode|SSE interface to avoid side

reactions and SCL formation, and promote fast Li⁺ transport through the cathode|SSE interface.

When a sulfide SSE is in contact with an oxide cathode material with a low Li⁺ chemical potential ($\mu_{Li}$) vs. $S^{2-}/S$, the sulfide SSEs will be oxidized even under open circuit voltage conditions, which will further promote structural degradations of SSEs and cathode active materials. As presented in Fig. 1a, an interphase forms between an oxide cathode material and a sulfide SSE with depleted Li⁺ on the sulfide electrolyte side, side reaction products as well as self-decomposition products, leading to a huge interfacial resistance. When an artificial oxide interlayer is constructed, the cathode|SSE interface can be considered as a combination of two interfaces, the cathode|interlayer and interlayer|SSE interfaces (Fig. 1b). Although the artificial interlayer can alleviate the Li⁺ redistribution compared to that shown in Fig. 1a, the Li⁺ depleted layer still exists. In addition, as delithiation/lithiation of the cathode during cycling is coupled with Li⁺ diffusion, it is highly dependent on parameters such as Li⁺ concentration and local potential. In this regard, the artificial oxide interlayer can not effectively mitigate the nonuniform distribution of Li⁺ concentration and electrochemical potential. Based on the idea that the high structural and

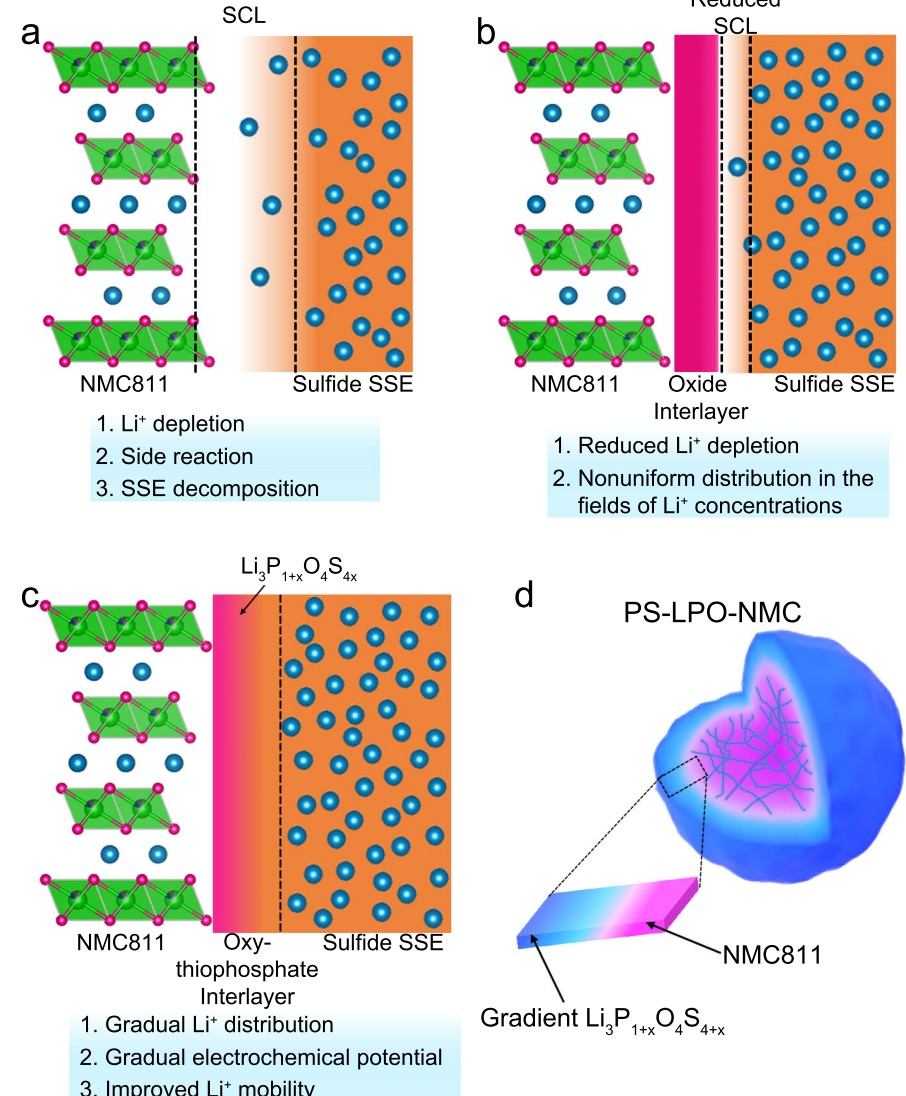

**Fig. 1 | Schematic illustration of the different types of interphases between NMC811 cathode and a sulfide SSE. a** Formation of thick SCL when uncoated NMC811 is in direct contact with a sulfide SSE; **b** reduced SCL with an oxide coating on NMC811; and **c** a gradient lithium oxide-oxy-thiophosphate interface tailoring a smooth transition from oxide-favored to sulfide-favored. **d** Schematic representation of an NMC811 primary particle (pink colour) with an ionic conductive and gradient $Li_3P_{1+x}O_4S_{4x}$ coating (blue colour). SCL stands for space-charge layer and PS-LPO-NMC stands for gradient $Li_3P_{1+x}O_4S_{4x}$-coated NMC811.

chemical similarities can reduce the interfacial resistance, a gradual lithium oxy-thiophosphate ($Li_3P_{1+x}O_4S_{4x}$) interface (Fig. 1c) is designed to simultaneously ensure homogeneous $Li^+$ diffusion, and avoid the SCL formation. Also, the gradient $Li_3P_{1+x}O_4S_{4x}$ interface can guarantee a higher $\mu_{Li}$ near the region in contact with sulfide SSE, avoiding the oxidation and decomposition of sulfide SSE. Benefiting from the gradient $Li^+$ concentration, gradient electrochemical potential, and minimized interfacial resistance, fast and stable $Li^+$ migration between the cathode and SSE can also be ensured.

Herein, we present an approach by fabricating a gradient $Li_3P_{1+x}O_4S_{4x}$ artificial SSE interface on the surface and grain boundary of NMC811 primary particles (Fig. 1d) by ALD of $Li_3PO_4$ and subsequent sulfurization using a $P_4S_{16}$ assisted solid-liquid process. The $Li_3P_{1+x}O_4S_{4x}$ coating showed a progressive concentration gradient with an increased sulfur content towards the outer surface of the coating. Due to the ionic conducting but electronically insulating nature of this gradient artificial coating, the side reactions between sulfide SSE and NMC811 were hindered. In addition, the electrochemical and chemical stabilities between the $Li_3P_{1+x}O_4S_{4x}$ coating and sulfide SSEs were improved because of the chemical similarity between the outer $Li_3PS_4$-like chemistry and the sulfide SSEs compared to approaches using oxide coatings. More importantly, benefiting from the gradient $Li_3P_{1+x}O_4S_{4x}$ coating on the surface and grain boundary of primary NMC811 particles, the $Li^+$ ions can migrate smoothly across the NMC811 | $Li_3P_{1+x}O_4S_{4x}$ | sulfide SSE interface, which can ultimately suppress the structural transformation of NMC811 from the favorable layered $LiNi_{0.8}Co_{0.1}Mn_{0.1}O_2$ phase to the unfavorable rock-salt $Ni_{0.8}Co_{0.1}Mn_{0.1}O_2$ phase. As a result, ASSLBs using gradient $Li_3P_{1+x}O_4S_{4x}$ coated NMC811 achieved a high reversible capacity of ~160 mAh g$^{-1}$ at 0.089 mA cm$^{-2}$ at 25 ± 5 °C with good retention of 80% after 250 cycles when integrated with the commercial $Li_{10}GeP_2S_{12}$ SSE.

## Results and discussion

### Application of a concentration gradient strategy at the solid-state electrolyte cathode interface

Directly fabricating a highly ionic conductive dense thin film on the surface of cathode particles is a promising route to ensure fast $Li^+$ migration across the cathode|SSE interface and reliable positive electrode performance. Pioneering work has been reported with infused oxide protection into the grain boundaries in Ni-rich NMC particles[33]. However, a single-component coating/protection is not enough to compensate for the long-term stability requirements for both the oxide cathodes and the sulfide SSEs which are distinct in chemistry and $\mu_{Li}$. Alternatively, a gradient coating with tailored $\mu_{Li}$ across its depth should effectively facilitate smooth $Li^+$ transport through the cathode| SSE interface and the grain boundaries of the Ni-rich layered oxide particles, ensuring structural integrity for the positive electrode active material during repeated lithiation/delithiation cycles[34]. The ALD technique has long been considered an effective way to achieve a uniform and conformal oxide thin-film coating on the surface of the cathode primary particles, but it is still difficult to deposit a sulfide SSE with high $Li^+$ conductivity. The in situ gradual growth of a sulfide SSE on the ALD oxide coating based on the interfacial diffusion reaction between ALD oxide coating and sulfur-rich precursor can be a strategy to achieve a stable gradient sulfide-to-oxide coating with high $Li^+$ diffusivity. Therefore, an NMC811 cathode with an ionic conductive thin coating of gradient $Li_3P_{1+x}O_4S_{4x}$ compositions reaching from the surface to the grain boundaries of the NMC811 particles (denoted as PS-LPO-NMC811) is proposed (Fig. 1d). The gradient sulfurization process was realized by a controlled reaction between the preformed $Li_3PO_4$ coating (by ALD) and a $P_4S_{16}$ solution.

Firstly, a 10 nm $Li_3PO_4$ layer was formed on the commercial NMC811 cathode by the ALD approach[35]. Subsequently, the $Li_3PO_4$-coated NMC811 (denoted as LPO-NMC811) powders were added into a $P_4S_{16}$/DEGDME (DEGDME short for diethylene glycol dimethyl ether) solution and stirred for 1 h to proceed the sulfuration. The developed sulfur-rich phosphorus sulfide molecule of $P_4S_{16}$ (tetrahedral structure with six -P-S-S-P and four P = S bonds)[36] was chosen to chemically react with the ALD $Li_3PO_4$ coating to form an oxy-thiophosphate $Li_3P_{1+x}O_4S_{4x}$ outer shell due to the highly favourable O-S exchange. The bond dissociation energies are 597 kJ mol$^{-1}$ and 346 kJ mol$^{-1}$ for the P-O bond and P-S bond, respectively[37]. In addition to sulfur-rich $P_4S_{16}$, $P_2S_5$ (one of the common P-S species) was also used to calculate the chemical reaction energy with $Li_3PO_4$[38] (Supplementary Fig. 1). The reaction energy between $P_2S_5$ and $Li_3PO_4$ is 0 meV atom$^{-1}$, indicating that $Li_3PO_4$ can not be sulfurized by $P_2S_5$. Benefited from the electron-donating property of the sulfur-rich environment and S-S bridge bonds in the structure of the $P_4S_{16}$ molecule, a negative reaction energy (−40 meV atom$^{-1}$) between the $P_4S_{16}$ and $Li_3PO_4$ was obtained, suggesting possible spontaneous sulfuration of $Li_3PO_4$ by $P_4S_{16}$. Moreover, it has been reported that the highly $Li^+$ conductive $Li_3PS_4$ is the dominant lithiation product of $P_4S_{16}$[39], thus the $Li_3P_{1+x}O_4S_{4x}$ coating can afford fast $Li^+$ migration. It should be noted that the formula of $Li_3P_{1+x}O_4S_{4x}$ here is mainly to simplify the reaction between $P_4S_{16}$ and $Li_3PO_4$ ($\frac{x}{4} P_4S_{16} + Li_3PO_4 \rightarrow Li_3P_{1+x}O_4S_{4x}$) while not the specific composition. The degree of sulfurization can be controlled by adjusting the mass ratio of $P_4S_{16}$ to $Li_3PO_4$. Scanning electron microscopy (SEM) and energy dispersive spectroscopy (EDS) mapping images of uncoated NMC811, LPO-NMC811, and PS-LPO-NMC811 samples treated with different $P_4S_{16}$ contents and followed by different additional annealing temperatures are shown in Supplementary Fig. 2–6. The SEM and EDS mapping results indicated that the NMC811 secondary particles were successfully coated with LPO or PS-LPO species. The PS-LPO-NMC811 sample obtained from 1 wt.% $P_4S_{16}$ treatment showed a thin conformal coating with obvious P and S signals without altering the surface morphology of the NMC811 particles (Supplementary Fig. 4), whereas a higher $P_4S_{16}$ content (e.g. 5 wt.%) had led to thick and uneven surface film formation (Supplementary Fig. 5). When the amounts of $Li_3PO_4$ and $P_4S_{16}$ were unbalanced, the excess unreacted $P_4S_{16}$ molecules were deposited and accumulated on the surface of NMC811. Moreover, high annealing temperatures will prevent the formation of a uniform surface coating on the NMC811 particles (Supplementary Fig. 6). Therefore, the PS-LPO-NMC811 sample treated with 1 wt.% $P_4S_{16}$ without annealing was selected for further study unless otherwise noted.

To confirm the effectiveness of ALD and in situ sulfurization process in building a gradient $Li_3P_{1+x}O_4S_{4x}$ coating covering both the surface and grain boundaries of the NMC811 particles, the LPO-NMC811, and PS-LPO-NMC811 samples were investigated by high-angle annular dark-field-scanning transmission electron microscopy (HAADF-STEM). The HAADF-STEM micrographs of the LPO-NMC811 sample at different magnifications are shown in Supplementary Fig. 7. The ALD process of $Li_3PO_4$ coating did not alter the morphology of the NMC811 particles, so the LPO-NMC811 particles maintained the initial morphology of NMC811, showing secondary particles as aggregates of 300-500 nm-sized primary particles with clear grain boundaries and occasional gaps (Supplementary Fig. 7a). An ideal ALD process only deposits a conformal coating on the designated substrate without changing the properties of the substrate. The primary particles were high crystalline layered NMC811, showing lattice fringes under high-resolution transmission electron microscopy (HRTEM, Supplementary Fig. 7b–d). A thin amorphous $Li_3PO_4$ layer of several nanometers was observed along the grain boundaries and on the surface of the particle (Supplementary Fig. 7e, f). The EDS elemental mapping (Supplementary Fig. 8) of a randomly selected region reveals the distribution of P not only on the surface but also on the grain boundaries of the particles, confirming the successful formation of ALD-$Li_3PO_4$ coating even on the inner primary particles. The uniform coating on the surface and grain boundaries by the ALD technique shows advantages in terms of precise and conformal coverage.

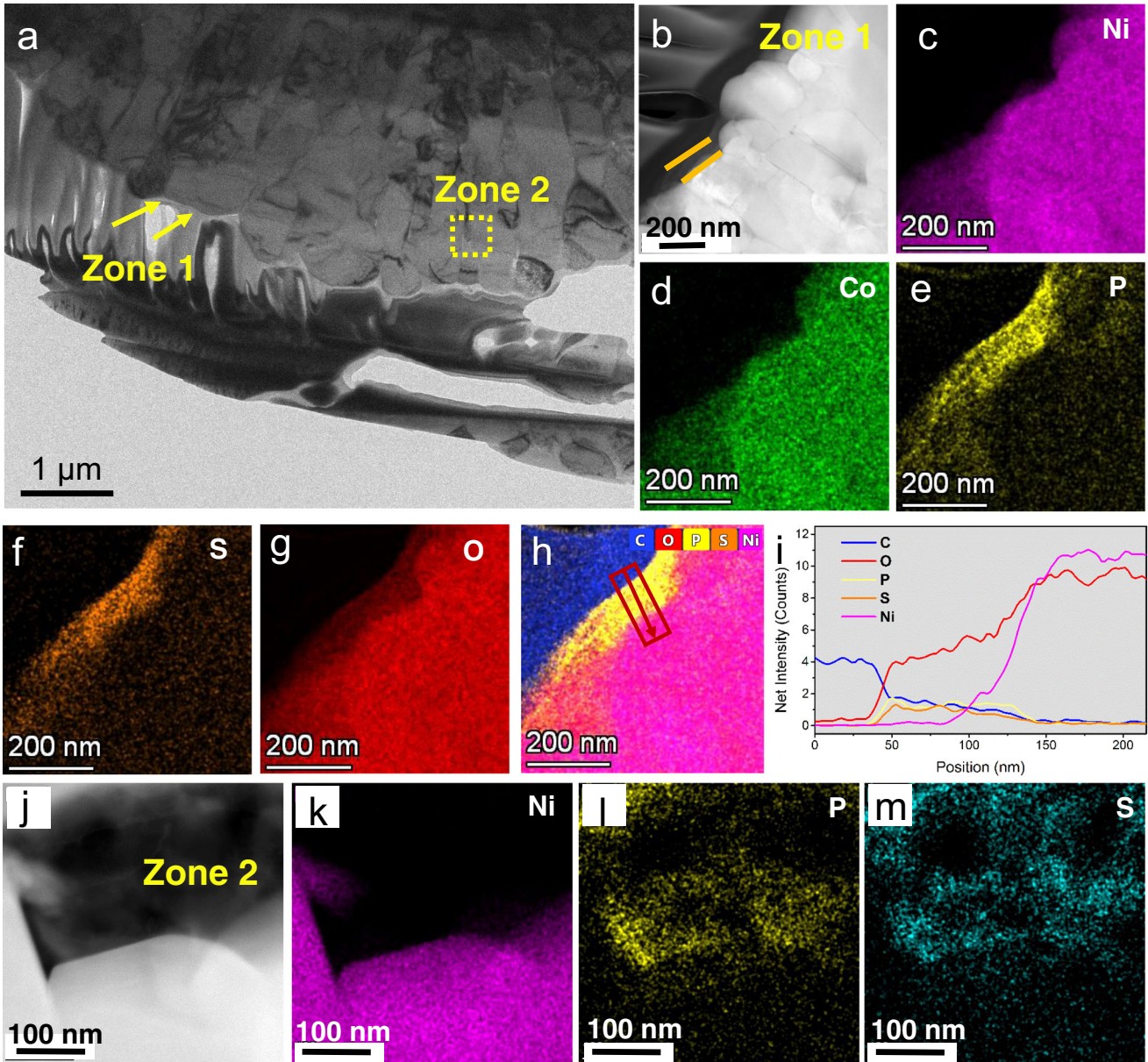

**Fig. 2 | TEM measurements of Li$_3$P$_{1+x}$O$_4$S$_{4x}$-coated NCM811 particles before electrode manufacturing. a** A low magnification TEM image of the PS-LPO-NMC811 secondary particle. **b** A HAADF-STEM image for Zone 1 in **a** focusing on the surface of the particle of PS-LPO-NMC811; and **c**–**h** the corresponding EDS elemental mapping of Ni, Co, P, S, O elements and the overlay map of C, O, P, S, Ni elements. **i** An EDS line scan across the region as marked in **h**. **j** A HAADF-STEM image of the inner primary particles (Zone 2 in **a**) of the PS-LPO-NMC811 sample; and (**k**–**m**) the corresponding EDS elemental mapping of Ni, P, S.

Figure 2 presents HAADF-STEM results and elemental mapping for the PS-LPO-NMC811 sample, showing a similar interior morphology to the LPO-NMC sample with observable grain boundaries and occasional gaps in a secondary particle. The cross-section TEM sample was obtained by the focused ion beam. The in situ sulfurization process based on a controlled reaction between P$_4$S$_{16}$ and the ALD-Li$_3$PO$_4$ coating led to a uniform and conformal thin Li$_3$P$_{1+x}$O$_4$S$_{4x}$ coating to the primary particle level. Due to the good penetration and diffusion of P$_4$S$_{16}$/DEGDME solution into the LPO-NMC811 particles, the P$_4$S$_{16}$ molecules can efficiently react with the Li$_3$PO$_4$ layer, in situ forming the desired Li$_3$P$_{1+x}$O$_4$S$_{4x}$ coating (Supplementary Fig. 9). Two zones (Fig. 2a) of the PS-LPO-NMC811 secondary particle were selected for further characterization. EDS elemental mapping of the surface (Zone 1, Fig. 2b–h) region revealed the distribution of P and S. The electron energy loss spectroscopy (EELS) line scans were further performed across the surface (Zone 1, Fig. 2h, i) and inner

grain boundaries and gaps (Zone 2, Supplementary Fig. 10h–j) regions to obtain the depth profiles of P, S, O, and other elements of interest for the PS-LPO-NMC sample. The S element shows a gradient distribution with a higher content at the surface and lower contents inwards. The P elemental distribution is relatively constant across the coating thickness. Both Zone 1 and Zone 2 reveal similar gradient distribution curves of the elements. The P and S elemental distributions are also revealed by the EDS elemental mapping of the inner grain boundaries and gaps (Zone 2, Fig. 2j–m and Supplementary Fig. 10). The EDS results confirm the in situ formation of gradient Li$_3$P$_{1+x}$O$_4$S$_{4x}$ interface on both the surface and grain boundaries of the NMC811 particles. The compositions at the grain boundary might be slightly different from that of the surface due to the possible diffusion of the transition metal ions inside the NMC811 particles. Supplementary Fig. 11 reveals the morphology of the Li$_3$P$_{1+x}$O$_4$S$_{4x}$ coated PS-LPO-NMC811 particles by HRTEM. The Li$_3$P$_{1+x}$O$_4$S$_{4x}$ layer is

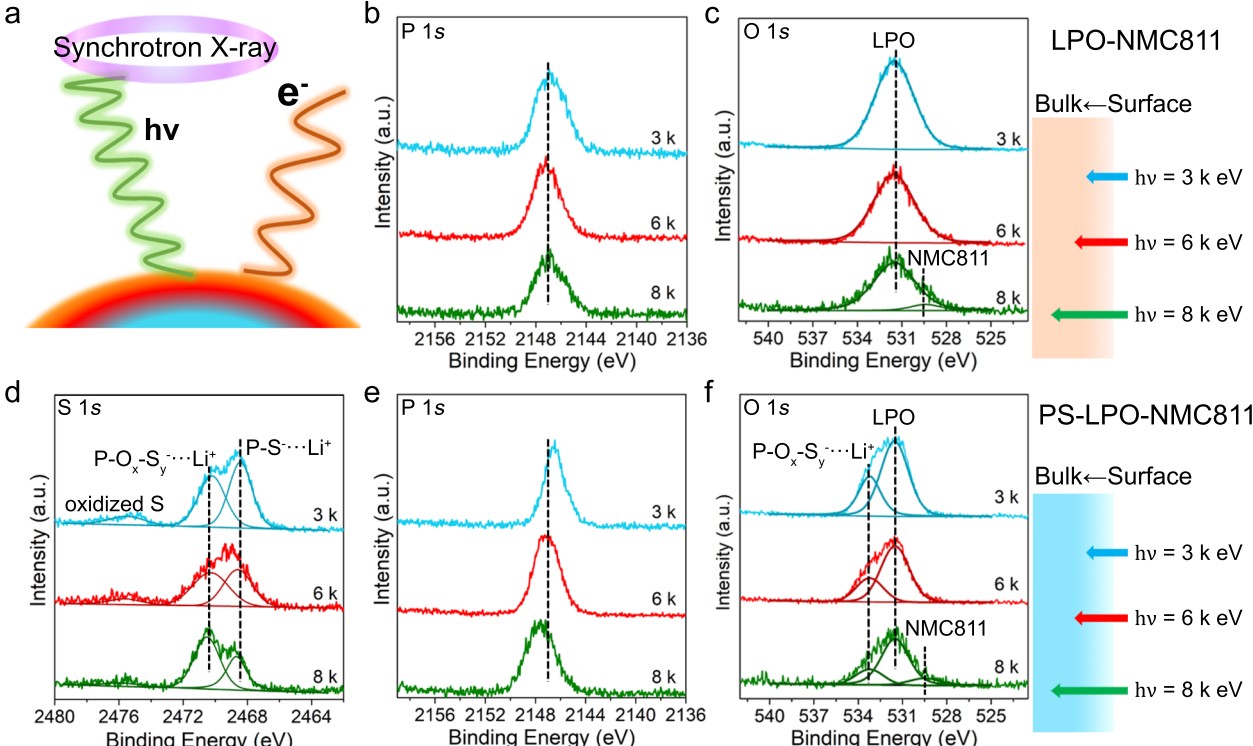

**Fig. 3 | Depth profiling of the LPO-NMC811 and the PS-LPO-NMC811 samples.** **a** Schematic of synchrotron-based high energy XPS with tunable energy. **b** P 1*s* and **c** O 1*s* HEXPS spectra of the LPO-NMC811 sample at different photon energies of 3000 (blue lines), 6000 (red lines), and 8000 (green lines) eV. **d** S 1*s*, **e** P 1*s*, and **f** O 1*s* HEXPS spectra of the PS-LPO-NMC811 sample at different photon energies of 3000 (blue lines), 6000 (red lines), and 8000 (green lines) eV.

10–20 nm in thickness with some crystalline clusters embedded in the major amorphous phase. The clusters possess a similar crystalline face to the Li-argyrodite phase of $Li_7PS_6$. Although the exact phase and composition of the clusters cannot be precisely determined, it proves that the reaction between $Li_3PO_4$ and $P_4S_{16}$ can originate compounds with similar Li-argyrodite phase clusters. The Li-argyrodite phase is a highly $Li^+$ conductive phase, so the $Li_3P_{1+x}O_4S_{4x}$ coating should possess a much higher $Li^+$ conductivity than the ALD-LPO layer or the bulk NMC811, ensuring the uniform and fast $Li^+$ flow on the surface and grain boundaries of NMC811 during charging and discharging processes. Therefore, problems caused by non-uniform $Li^+$ flow during cycling (e.g. transition metal ions diffusion, undesired structure degradation of layered-to-spinel transformation, and the buckling stress) can be alleviated[21]. In addition, the mechanical analysis based on atomic force microscopy measurements (Supplementary Fig. 12) proves the low Young's modulus of the $Li_3P_{1+x}O_4S_{4x}$ coating layer, which is propitious to achieving conformal contact with NMC811 particles. Tailoring the structure and chemistry of the surface and grain boundaries for NMC811 through a stable and highly $Li^+$ conductive coating, such as the gradient $Li_3P_{1+x}O_4S_{4x}$ shown here, can thus enhance the cathode performance.

In addition to the morphological, structural, and elemental distribution results provided by HRTEM, the synchrotron X-ray absorption near edge structure (XANES), high energy X-ray photoelectron spectroscopy (HEXPS), and Time-of-Flight secondary ion mass spectrometry (TOF-SIMS) measurements and analyses were performed to obtain chemical information for the $Li_3P_{1+x}O_4S_{4x}$ coating. The XANES spectra of S and P K-edges for the PS-LPO-NMC811 sample (Supplementary Fig. 13a, b) reveal that the chemistry of S and P in the $Li_3P_{1+x}O_4S_{4x}$ coating is different from that in the ALD $Li_3PO_4$ on the LPO-NMC811 sample, $Li_3PS_4$, or $P_4S_{16}$. Although the exact products from the reaction between $P_4S_{16}$ and $Li_3PO_4$ layer are unclear, the formation of the Li-P-O-S containing layer occurred via interfacial inter-

diffusion of elements due to the different fields of $Li^+$ concentrations and chemical potentials of P and S within the layer. On the other hand, the similar XANES spectra of Ni, Mn, and Co K-edges for uncoated NMC811, LPO-NMC811, and PS-LPO-NMC811 samples (Supplementary Fig. 13c, d) demonstrate the unchanged chemical state of the NMC811 cathode after the ALD coating process or/and the in situ sulfurization process. The chemical composition of the surface layers was measured by XPS as presented in Supplementary Figs. 14–16. A single spin-orbit doublet with peaks at binding energies of 133.67 ($2p3/2$) and 134.51 eV ($2p1/2$) is observed in the P $2p$ spectrum for the LPO-NMC811 sample[40]. Notably, the doublet shifts to lower energies of 133.14 and 133.98 eV for the PS-LPO-NMC811 sample. The relatively low energy shift for P $2p$ is reasonable considering the strong binding between $P^{5+}$ (hard acid) and $O^{2-}$ (hard base) compared to the case when a soft base of $S^{2-}$ was partially involved for the PS-LPO-NMC811 sample. The S $2p$ spectra are more complicated. Three spin-orbit doublets at binding energies of 161.2/162.36, 162.9/164.06, and 166.4/167.56 eV that being characteristic of S to $PS_4^{3-}$, oxy-thiophosphate species ($P-O_x-S_y^-...Li^+$), and oxidized sulfur species ($SO_3^{2-}$), respectively, can be deconvoluted for the S $2p$ spectrum of the PS-LPO-NMC811[41,42]. The XPS results demonstrate the successful coating of $Li_3PO_4$ on NMC811 by ALD approach and followed sulfurization of LPO-NMC811 to PS-LPO-NMC811 by $P_4S_{16}$. As consistent with the XANES results, the Ni $2p$, Mn $2p$, and Co $2p$ XPS spectra of the uncoated NMC811, LPO-NMC811, and PS-LPO-NMC811 indicate good preservation of the NMC811 chemistry after ALD and sulfuration processes (Supplementary Fig. 16).

Nondestructive depth profiling analyses were performed by synchrotron-based HEXPS using the soft X-ray microcharacterization beamline (SXRMB). As presented in Fig. 3a, the probing depth of the photoelectron emission can be tuned by varying the beam energy, thus providing chemical states and elemental composition distribution information across a thickness for interfacial materials[43–45]. The present measurements were performed at 3000, 6000, and 8000 eV of photon energies to probe the different depths and compare the

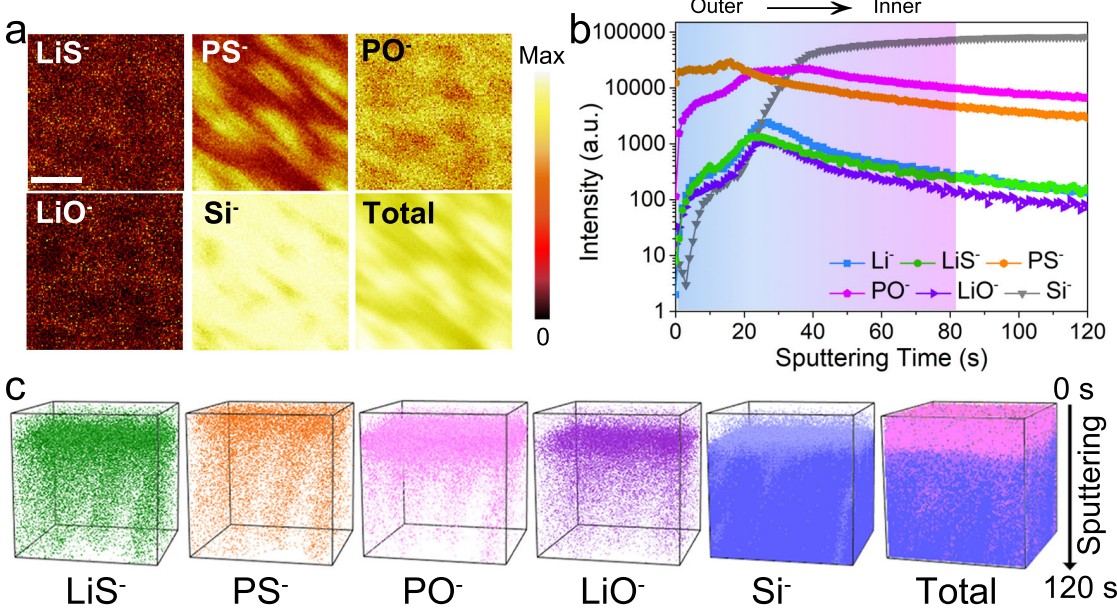

**Fig. 4 | Chemical compositions of the gradient oxy-thiophosphate thin film.**
**a** TOF-SIMS secondary ion images of LiS⁻, PS⁻, PO⁻, LiO⁻, and Si⁻ species after Cs⁺ consecutive sputtering for 120 s (the length of scale bar is 40 μm) for the PS-LPO-Si sample. **b** Depth profile of various secondary ion species obtained by sputtering.

**c** The 3D view images of the sputtered volume corresponding to the depth profiles in (**a**) show the gradient oxy-thiophosphate distribution. The analysis area is 75 × 75 μm².

coating compositions of the LPO-NMC811 and the PS-LPO-NMC811 samples. The binding energy in all the spectra was calibrated using a pure Au foil. The main peaks of P 1$s$ and O 1$s$ HEXPS of LPO-NMC811 (Fig. 3b, c) are located at 2147.1 eV and 531.5 eV, respectively, which corresponded to the phosphorus and oxygen in phosphates[35,40,46]. The weak signal of O 1$s$ at 529.5 eV captured at high photon energy (8000 eV) is attributed to the bulk NMC811[47]. Figure 3d–f shows the depth-resolved HEXPS spectra of S 1$s$, P 1$s$, and O 1$s$ core levels collected for the PS-LPO-NMC811 sample at different photon energies. In addition to the small peak around 2475.6 eV assigned to oxidized sulfur species, another two peaks are observed at 2470.2 and 2468.6 eV. The peak at lower binding energy can be assigned to thiophosphate (P-S⁻...Li⁺) and the other peak at 2470.2 eV is attributed to oxy-thiophosphate species (P-O$_x$-S$_y$⁻...Li⁺)[48]. The peak area ratios of thiophosphate/oxy-thiophosphate are summarized in Supplementary Table 1. By comparison, the oxy-thiophosphate peak becomes dominant over the thiophosphate peak with increasing photon energy, indicating a higher oxy-thiophosphate content towards the inner depth of the Li₃P₁₊ₓO₄S₄ₓ coating of the PS-LPO-NMC811 sample. Both thiophosphate and oxy-thiophosphate species should be originated from the sulfurization of the Li₃PO₄ through the reaction between ALD-Li₃PO₄ and P₄S₁₆.

Different from the separated S 1$s$ peaks, the P 1$s$ XPS spectra (Fig. 3e) of the PS-LPO-NMC811 obtained at various photon energies exhibit only one peak. Deconvolution of the peak could be difficult because of the minor difference (-0.2 eV) in the P 1$s$ binding energy between the thiophosphate and oxy-thiophosphate species[46]. However, the apparent shift of the peak to low binding energies along with the decreasing of photon energy can be observed, implying varied S-to-O atomic ratio in the oxy-thiophosphate species. The P 1$s$ binding energy in the oxy-thiophosphate species can be higher with an increasing O content because the bonding between strong Lewis acid of P⁵⁺ and strong Lewis based of O²⁻ is relatively stronger than the bonding with S²⁻. The formation of oxy-thiophosphate species (P-O$_x$-S$_y$⁻...Li⁺) is also evident by the peak at 533.2 eV in the O 1$s$ XPS spectra

(Fig. 3f)[42]. The relative decrease in intensity of the oxy-thiophosphate peak upon increasing photon energy (Supplementary Table 2) indicates the higher O content of Li-P-O-S species in the inner surface layer. Overall, the trends observed for the P 1$s$ and O 1$s$ XPS spectra are well consistent with the intensity ratio reflected from the S 1$s$ XPS spectra, further verifying the gradient oxy-thiophosphate distribution across the coating thickness on the PS-LPO-NMC811 sample.

TOF-SIMS was further performed to identify the chemical composition and elemental depth distributions of the gradient Li₃P₁₊ₓO₄S₄ₓ artificial interlayer (Fig. 4). To avoid the influences of uneven surfaces of the NMC811 particles, the analysis was conducted with gradient oxy-thiophosphate thin film on a silicon wafer substrate that underwent the same treatment process (ALD coated LPO followed by reaction with a P₄S₁₆ solution). From the chemical ion images (Fig. 4a), species of LiS⁻, PS⁻, PO⁻, and LiO⁻ are observed and indicate the deposition of the gradient oxy-thiophosphate thin film. The LiS⁻ and PS⁻ are traced as the characteristic components for the sulfurized-Li₃PO₄ that contains thiophosphate (P-S⁻...Li⁺) and oxy-thiophosphate species (P-O$_x$-S$_y$⁻...Li⁺), whereas the PO⁻ and LiO⁻ are originated from the Li₃PO₄. In Fig. 4b, the signals of LiS⁻ and PS⁻ gradually increase in the initial ~20 s of sputtering, corresponding to the sulfurized-Li₃PO₄ in the outer surface layer. Afterward, these signals significantly decrease along with the sputtering time. In contrast, the intensity of the PO⁻ signal from Li₃PO₄ is relatively stable until sputtering for ~80 s. The distributions of those species are visualized as 3D render images shown in Fig. 4c. The signals of LiS⁻ and PS⁻ species are relatively higher towards the outer surface, especially the PS⁻ distribution. On the other hand, the PO⁻ and LiO⁻ signals are distributed mainly below the sulfurized layer and close to the Si substrate. Moreover, the intensities of those species gradually change along with the depth. Thus, the TOF-SIMS results provide strong evidence that the oxy-thiophosphate protective gradient interlayer design with controllable composition was successfully realized via the ALD and followed the partial sulfurization process. As confirmed by the HEXPS and TOF-SIMS analyses, the outer surface of the gradient oxy-thiophosphate thin film is rich in S content for the

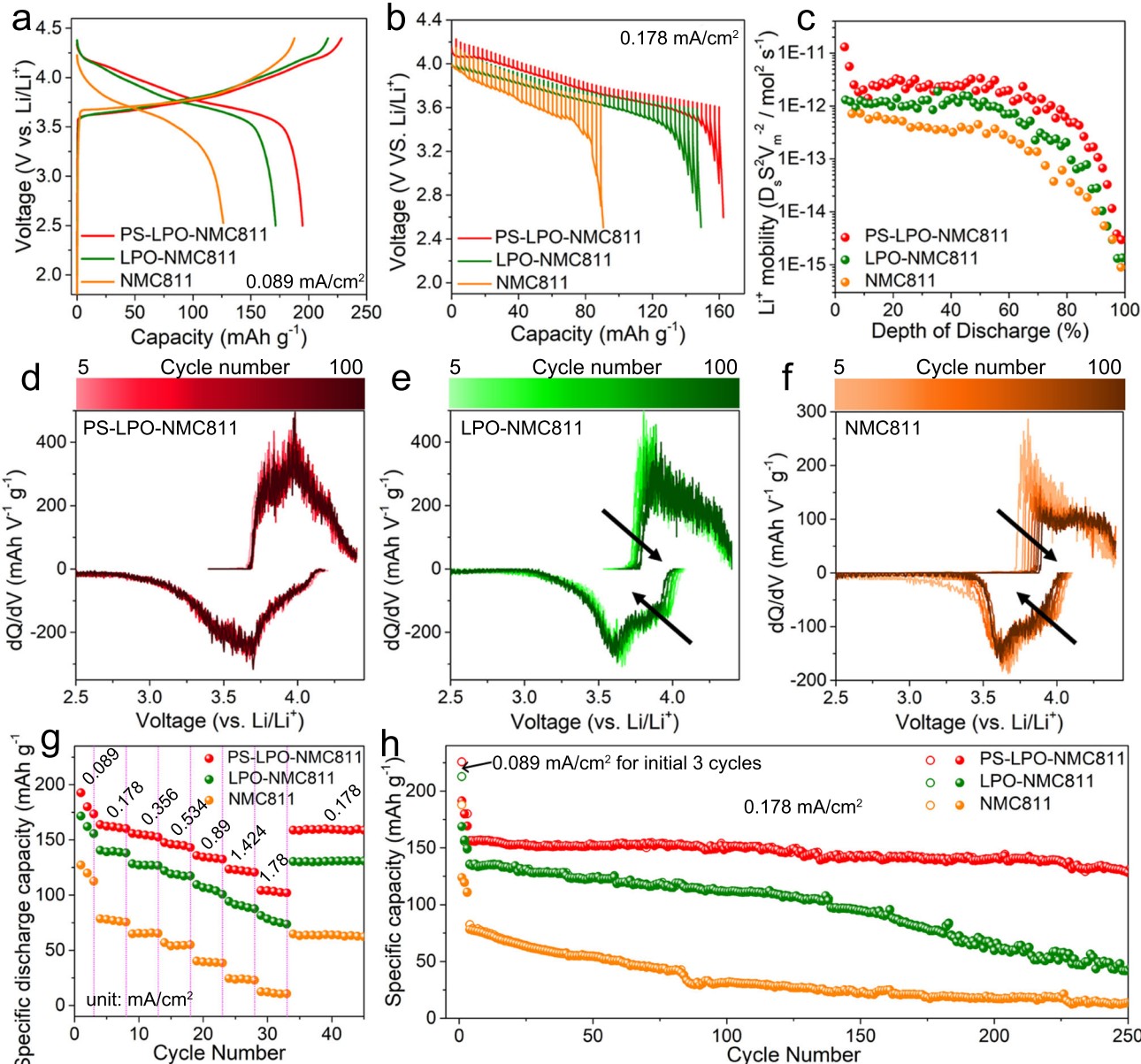

**Fig. 5 | Electrochemical performance of all-solid-state Li-ion cells with an In anode, $Li_{10}GeP_2S_{12}$ solid electrolyte and various NCM811 cathodes at 25 °C.** **a** Charge/discharge curves of the first cycle at 0.089 mA cm⁻², **b** GITT curves during the discharge process, and **c** corresponding Li⁺ diffusion coefficients of the three kinds of NCM811 cathodes during different discharge states. The error bar of GITT data is around 2%, which is original from the mechanism error bar in this testing (±0.5 μA) and the data analysis (±0.1 mV). **d**–**f** The dQ/dV curves of the three kinds of NCM811 cathodes during the initial 100 cycles at 0.178 mA cm⁻². **g** Rate capabilities and **h** cycling performance of the three kinds of NCM811 cathodes. Red for the PS-LPO-NMC811 cathode, green for the LPO-NMC811 cathode, and orange for the NMC811 cathode.

sulfurized-$Li_3PO_4$ species mainly including thiophosphate (P-S⁻...Li⁺) and oxy-thiophosphate (P-O$_x$-S$_y$⁻...Li⁺), and the inner surface is rich in O content resembling the pristine $Li_3PO_4$ species. The distribution of components varies with the depth of the $Li_3P_{1+x}O_4S_{4x}$ artificial interlayer. By design, the high structural/component similarity of the sulfur-rich outer coating to the sulfide SSEs and gradient Li⁺ concentration and electrochemical potential towards the cathode surface can ensure a good cathode|SSE interface for ASSLBs.

**Battery testing of the gradient oxy-thiophosphate-coated NCM811 cathodes in all-solid-state lithium-ion cell configuration**
The different types of NCM811 cathodes were mixed with the commercial $Li_{10}GeP_2S_{12}$ (LGPS) SSE as the cathode composites. ASSLBs using each cathode composite, LGPS SSE, and an In anode were assembled in homemade KP-Solid cells (Supplementary Fig. 17).

Figure 5a depicts the representative charge/discharge voltage profiles of the three ASSLBs at 0.089 mA cm⁻² at 25 ± 5 °C. The initial charge and discharge capacities of 187.7 and 126.4 mAh g⁻¹, respectively, were demonstrated for uncoated NCM811, leading to an initial Coulombic efficiency of 67.3%. The initial charge/discharge capacities were increased to 216.7 and 171.4 mAh g⁻¹ (79.1% Coulombic efficiency) for the LPO-NMC811 cathode. The PS-LPO-NMC811 cathode after in situ sulfurization achieved a further increase in specific capacities to 228.5 and 194.7 mAh g⁻¹ (85.2% Coulombic efficiency). The severe Li loss due to the side reactions between the sulfide SSE and uncoated NCM811 was proved by its oxidation potential before reaching the delithiation potential of the NCM811 cathode as shown at the beginning of the initial charge process (Supplementary Fig. 18)[49]. In contrast, no such behavior was observed for the PS-LPO-NMC811 cathode. Moreover, the electrochemical impedance spectroscopy (EIS) measurement and

analyses of the PS-LPO-NMC811 cathode at different charge/discharge states (Supplementary Fig. 19 and Supplementary Table 3) showed small interfacial resistance change, demonstrating that the gradual $Li_3P_{1+x}O_4S_{4x}$ coating can reduce the SCL layer and side reactions between NMC811 and sulfide SSEs.

The galvanostatic intermittent titration technique (GITT) was employed to probe the $Li^+$ dynamics at different charge states for the three NMC811 cathodes in ASSLBs. The corresponding open-circuit voltage (OCV) profiles and polarization voltages during discharge at 0.178 mA cm$^{-2}$ are presented in Fig. 5b and Supplementary Fig. 20. The extracted results on lithium mobility are summarized in Fig. 5c. The calculation of diffusion coefficient of $Li^+$ ($D_{Li+}$) can be calculated based on Eq. (1),

$$D_{Li+} = \frac{4}{\pi\tau}\left(\frac{m_{NMC811}V_m}{M_{NMC811}S}\right)^2\left(\frac{\triangle E_s}{\triangle E_\tau}\right)^2 \tag{1}$$

where $m_{NMC811}$ is the mass of NMC811 in the cathode composite, $V_m$ is the molar volume of NMC811, $\tau$ is the relaxation time (2 h), S is the active area of the composite electrode, $M_{NMC811}$ is the molar mass of NMC811, $\triangle E_s$ and $\triangle E_\tau$ is the steady-voltage change after the relaxation and the transient-voltage change after 10 min discharge process at 0.178 mA cm$^{-2}$, respectively. Since the values of molar volume and active area can not be obtained accurately, the $Li^+$ mobilities in Fig. 5c were normalized to $D_{Li+}S^2V_m^{-2}$ to compare the relative $Li^+$ dynamics within the three NMC811 cathodes. The PS-LPO-NMC811 cathode presents the smallest polarization potential and the highest normalized $D_{Li+}S^2V_m^{-2}$ value during the entire discharge process, indicating its fastest $Li^+$ dynamics. The fast $Li^+$ migration for the PS-LPO-NMC811 cathode can be attributed to three reasons: (1) the significantly reduced SCL formation, (2) the minimized $Li^+$ migration barrier ensured by the gradual $Li^+$ concentration and electrochemical potential, and (3) the intrinsically high $Li^+$ conductivity of the $Li_3P_{1+x}O_4S_{4x}$ coating with full coverage on the surface and grain boundaries for the NMC811 particles.

The electrochemical reversibility of the three NMC811 cathodes was further evaluated by the differential capacity analysis curves during the initial 100 cycles as shown in Fig. 5d–f. The peaks which indicate the charge/discharge overpotentials essentially remain unchanged for the PS-LPO-NMC811 cathode; the peaks of the LPO-NMC811cell show minor shifts; in contrast, the uncoated NMC811 cathode exhibits a high-voltage shift for the anodic peak and low-voltage shift for the cathodic peak with the increasing cycling number. The significant voltage fading for the uncoated NMC811 cathode indicates the presence of side reactions between uncoated NMC811 and sulfide SSE. The negligible polarization change of the PS-LPO-NMC811 cathode should be ascribed to the designed $Li_3P_{1+x}O_4S_{4x}$ interlayer, as discussed above, which can endow both stable interface and high $Li^+$ migration. Figure 5g compares the rate performances of the three NMC811 cathodes from 0.089 to 1.78 mA cm$^{-2}$. The PS-LPO-NMC811 cathode exhibits a high capacity of 103 mAh g$^{-1}$ at 1.78 mA cm$^{-2}$, however, the LPO-NMC811 and uncoated NMC811 cathodes show very low capacities of 75 and 12 mAh g$^{-1}$, respectively. The long-term cycling stability and corresponding Coulombic efficiencies of the three NMC811 cathodes are presented in Fig. 5h and Supplementary Fig. 21, with 0.089 mA cm$^{-2}$ for the initial three cycles and 0.178 mA cm$^{-2}$ for the following cycles. The PS-LPO-NMC811 cathode demonstrates a high discharge capacity of 161 mAh g$^{-1}$ at the 4th cycle and retained 128 mAh g$^{-1}$ after 250 cycles (80% capacity retention). In contrast, both the LPO-NMC811 and the uncoated NMC811 cathodes exhibit severe capacity decay upon cycling, leading to 31% and 15% capacity retentions after 250 cycles, respectively. Moreover, the electrochemical performance of the PS-LPO-NMC811 cathodes with excess sulfurization (for example, treated by 2.5 wt.% and 5 wt.% of $P_4S_{16}$) show drawbacks as well

(Supplementary Fig. 22). Therefore, the optimized $Li_3P_{1+x}O_4S_{4x}$ coating plays a vital role in the good performance of relative ASSLBs.

## Postmortem ex situ positive electrode microscopy measurements and analyses

HAADF-STEM micrographs were obtained to distinguish the morphology and the structural transformation of the grain boundary on the LPO-NMC811 and PS-LPO-NMC811 particles after long-term cycling (Figs. 6, 7). Figure 6a depicts the cross-sectional image of LPO-NMC811 particles after 100 cycles of charge/discharge at 0.178 mA cm$^{-2}$. The particle size didn't change significantly, and neither pulverization nor aggregation is observed. The shape of the primary particles remains similar to the initial LPO-NMC811 particles before and after cycling (Supplementary Fig. 7a). However, the grain boundaries appeared to thicken significantly, which include a 30-50 nm thickness of the structurally transformed layer after cycling (Fig. 6b). Figure 6c–f gives the high-resolution STEM investigations of the phase boundaries in Fig. 6b. The structural transformation from the layered $LiNi_{0.8}Co_{0.1}Mn_{0.1}O_2$ (Fig. 6d) to a spinel-like phase $Li(Ni_{0.8}Co_{0.1}Mn_{0.1})_2O_4$ (Fig. 6e) through the diffusion of transition metal (TM) atoms (Fig. 6f) have been found. The formation of a spinel-like phase caused structural distortions that may have led to the capacity fading and increased overpotentials upon prolonged cycling, as previously discussed for the LPO-NMC811-based ASSLBs (Fig. 5e). This was due to the redox activity of spinel moieties around 3 V (vs. $Li^+$/Li) corresponding to the $Mn^{4+}$/$Mn^{3+}$ couple, in addition to their major redox activity of layered $LiNi_{0.8}Co_{0.1}Mn_{0.1}O_2$ at higher potentials (around 4.2 V, vs. $Li^+$/Li). The formation of spinel structure at the grain boundaries will be continuously promoted by the diffusion of TM atoms originating from the different chemical potentials at the interface. Although the EDS mapping still shows the existent $Li_3PO_4$ layer on the surface of the particles after cycling (Fig. 6g and, Supplementary Fig. 23), this layer is not sufficient to protect the NMC811 particles for long-term cycling. The formation of the spinel phase at the surface and grain boundaries has been identified as one of the major structural degradation mechanisms for cathode failure during battery cycling[50]. Surprisingly, these layered-to-spinel transformation behaviors did not occur in the PS-LPO-NMC811 particles (Fig. 7). Apparently, the gradual $Li_3P_{1+x}O_4S_{4x}$ artificial layer in the grain boundaries of the particles prevented the formation and growth of the spinel phase. The NMC811 maintained a favorable layered structure even after long-term cycling (Fig. 7b–e). There is only a thin rock-salt phase $Ni_{0.8}Co_{0.1}Mn_{0.1}O_2$ formation on the outer surface of the primary particle (Fig. 7c and, Supplementary Fig. 24). The thickness of the rock-salt phase is only ~2 nm (Fig. 7b). As consistent with the structural differences of the grain boundaries described above, the chemical differences between LPO-NMC811 and PS-LPO-NMC811 after cycling reflect a distinctively different diffusion process and protection effects of the two coatings. Therefore, the sustainability of NMC811 grain boundaries can be used to evaluate the stability of NMC811 particles during the lithiation-delithiation process and the uniformity of $Li^+$ migration across any interface. The EDS mapping shows that the $Li_3P_{1+x}O_4S_{4x}$ coating homogeneously occurred at the grain boundaries even after long-term cycling (Fig. 7e). This artificial gradient SSE coating layer is effective in protecting the Ni-rich NMC811 layered structure for good cycling stability.

In summary, a thin and gradient $Li_3P_{1+x}O_4S_{4x}$ coating was proposed and successfully synthesized to tackle the poor cycling stability of the high-capacity Ni-rich NMC811 cathode materials for sulfide-based ASSLBs. The highly ionic conductive and gradual $Li_3P_{1+x}O_4S_{4x}$ coating was fabricated with full coverage on the surface and grain boundary of the primary NMC811 particles by ALD-formed $Li_3PO_4$ and subsequent in situ sulfurization. The $Li_3P_{1+x}O_4S_{4x}$ interface was 10–20 nm in thickness with some crystalline clusters embedded in the major amorphous phase. In-depth analyses of HRTEM, synchrotron-based HEXPS, and TOF-SIMS measurements confirmed the gradient

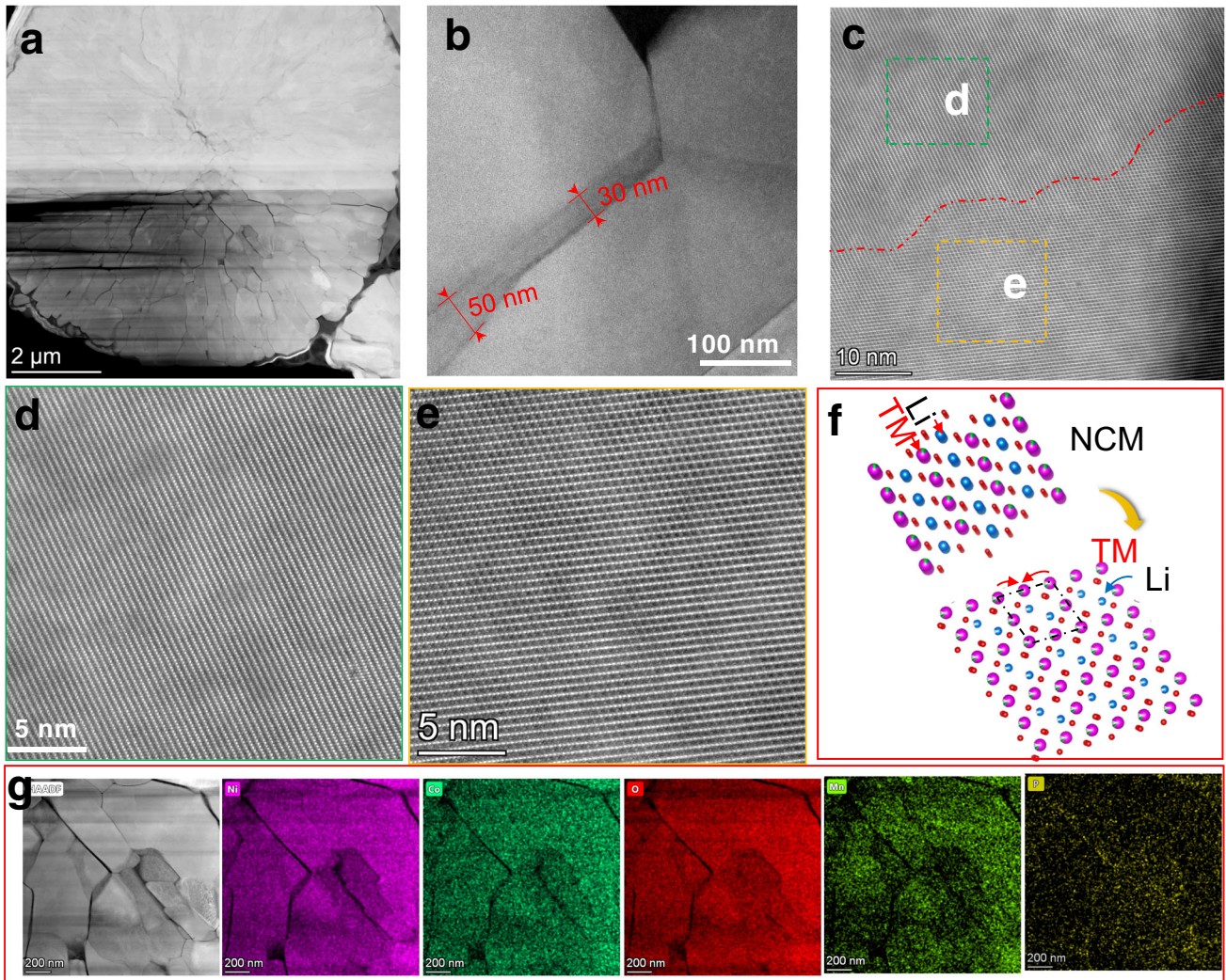

**Fig. 6 | Ex situ postmortem TEM measurements of cycled LPO-NMC811-based positive electrodes. a** Cross-sectional STEM image of the entire secondary particle. **b–f** High-resolution STEM images showing the layered-to-spinel transformation layer at the grain boundaries, the lattice fringes for the layered NMC structure, and the spinel phase at the boundaries after 100 cycles at fully discharged state of the In| LGPS | LPO-NMC811 cell. **g** The HAADF-STEM image and the corresponding elemental mapping for the LPO-NMC811 particles after 100 cycles at fully discharged state of the In|LGPS | LPO-NMC811 cell. The cells were cycled at 0.178 mA cm$^{-2}$ at 25 °C.

compositions involving S-rich Li-P-O-S species (thiophosphate P-S$^-$...Li$^+$ and oxy-thiophosphate P-O$_x$-S$_y$$^-$...Li$^+$) towards the outer surface and the O-rich Li-P-O-S species towards the inner cathode interface. Tailoring both the surface and grain boundary structure and chemistry by the gradient Li$_3$P$_{1+x}$O$_4$S$_{4x}$ coverage with stable and fast Li$^+$ transport across was demonstrated to dramatically reduce the structural degradation and the layered-to-spinel transformation at the grain boundary. Thus, the capacity retention and voltage stability of the cathode were significantly enhanced. The gradient interface enabled the In|LGPS | PS-LPO-NMC811 ASSLBs with highly stable cycling performance over 250 cycles with a specific discharge capacity retention of 80% (from the 4th to the 250th with an applied areal current of 0.178 mA/cm$^2$ at 25 °C.

## Methods

### Materials

Commercial LiNi$_{0.8}$Co$_{0.1}$Mn$_{0.1}$O$_2$ (NMC811, average primary around 10 um and secondary particle size around 500 nm, provided as not carbon-coated) electrode materials were purchased from China Automotive Battery Research Institute (China). Commercial Li$_{10}$GeP$_2$S$_{12}$ (LGPS) solid electrolyte powder (with less than 100 um size) was purchased from MSE Supplies LLC.

### Preparation of ALD Lithium phosphate coated NMC811 (LPO-NMC811) cathode

The lithium phosphate (LPO) was deposited on NMC811 cathode using lithium tertbutoxide (LiOtBu, 97%, Sigma-Aldrich) and trimethyl phosphate (TMPO, ≥ 99%, Sigma-Aldrich) as precursors with a deposition temperature of 250 °C in a Savannah 100 ALD system (Cambridge Nanotech, USA)[35]. The source temperatures for LiOtBu and TMPO were 180 °C and 75 °C, respectively. During one ALD cycle, LiOtBu and TMPO were alternatively introduced into the reaction chamber with a pulse time of 2 s, and the pulsing of each precursor was separated by a 15 s purge with N$_2$. The growth rate for the ALD LPO is ~0.7 nm/cycle.

### Preparation of P$_4$S$_{16}$ modified LPO-NMC811 (PS-LPO-NMC811) cathode

The gradient Li$_3$P$_{1+x}$O$_4$S$_{4x}$ coated NMC811 cathodes were obtained via in situ solution-based method. Firstly, 10 mg of P$_4$S$_{16}$ powders[36] were dissolved into 1 mL solvent of diethylene glycol dimethyl ether (DEGDME, 99.5%, Sigma-Aldrich) with the calculated concentration of 10 mg mL$^{-1}$. Secondly, 1 g of LPO-NMC811 powders were dispersed into the solution and stirred for 2 h at 25 °C. Then the final PS-LPO-NMC811 cathode was obtained after drying at 80 °C for 2 h to

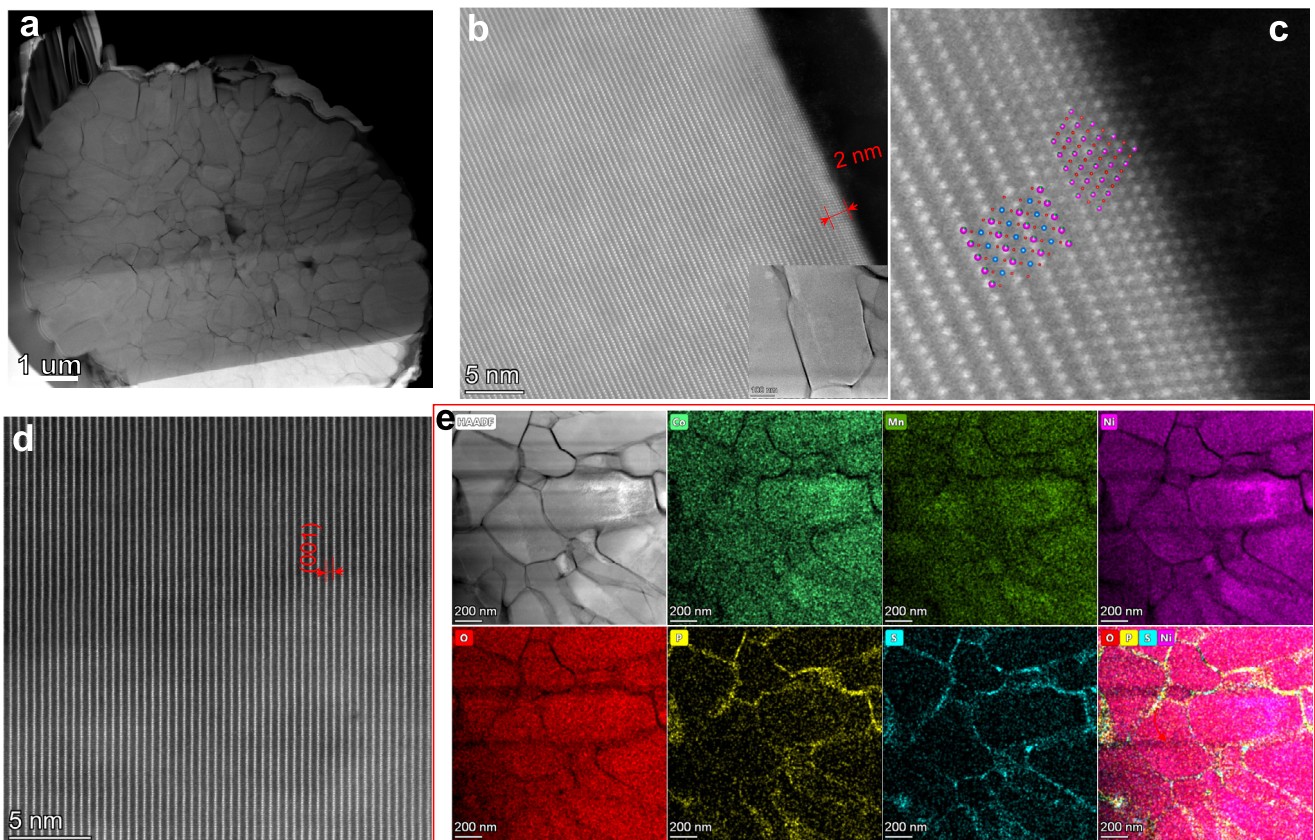

**Fig. 7 | Ex situ postmortem TEM measurements of cycled Li₃P₁₊ₓO₄S₄ₓ-coated NCM811-based positive electrodes. a** Cross-sectional STEM image. **b**–**d** The HR-STEM images for the layered NMC structure and the rock-salt phase at the primary particle surface after 100 cycles at a fully discharged state of the In|LGPS | PS-LPO-NMC811 cell. **e** The HADDF-STEM image and the corresponding elemental mapping from the PS-LPO-NMC particles after 100 cycles at a fully discharged state of the In| LGPS | PS-LPO-NMC811 cell. The cells were cycled at 0.178 mA cm⁻² at 25 °C.

remove the residual solvent of DEGDME inside the glovebox (inert atmosphere). The weight fraction of $P_4S_{16}$ in the resulting cathode powder was about 1%. Moreover, the thickness of the oxythiophosphate outer interlayer can be adjusted by changing the weight fraction of $P_4S_{16}$ to 2.5% and 5%, which can be adjusted the volume of 10 mg mL⁻¹ $P_4S_{16}$/DEGDME solution to 2.5 mL and 5 mL, respectively.

## Physicochemical characterizations

The chemical information of the uncoated-NMC811, LPO-NMC811, and PS-LPO-NMC811 samples was characterized by an X-ray photoelectron spectrometer (XPS, ESCALAB 250 spectrometer, Perkin-Elmer). The XPS spectra were fitted with Gaussian-Lorentzian functions and a Shirley-type background. The spin-orbit split peaks for P $2p$ ($2p_{1/2}$, $2p_{3/2}$) is constrained using a separation of 0.84 eV and the intensity ratio of $2p_{3/2}$:$2p_{1/2}$ about 1.192. The spin-orbit split peaks for S $2p$ ($2p_{1/2}$, $2p_{3/2}$) are constrained using a separation of 1.16 eV and the intensity ratio of $2p_{3/2}$:$2p_{1/2}$ about 1.677. The chemical information of the LPO-NMC811 and PS-LPO-NMC811 samples with deep profile was further tested by high-energy X-ray photoelectron spectroscopy (HEXPS), which was performed on the soft X-ray microcharacterization beamline (SXRMB) at the CLS30 under different energies. The X-ray absorption near edge structure (XANES) measurement was carried out at the Canadian Light Source (CLS). S, P, Ni, Co, and Mn K-edge XANES were collected using fluorescence yield mode on the soft X-ray microcharacterization beamline (SXRMB) at the CLS30. To avoid the air exposure effect, all the samples were covered with Mylar film in the glovebox under Ar, and then transferred to the chamber of the corresponding beamline. The morphologies of

various cathodes were observed using field emission scanning electron microscopy (FESEM, Hitachi S4800), high-resolution transmission electron microscopy (HRTEM, Talos F200), and High angle annular dark field-scanning transmission electron microscopy and energy-dispersive X-ray spectroscopy (EDS) (HAADF-STEM and EDS, Titan Themis Z 60-300). The TOF-SIMS measurements were conducted using a TOF-SIMS IV (ION-TOF GmbH, Germany) with a bismuth liquid metal ion source (25 keV). The base pressure in the analysis chamber is around 10⁻⁸ mbar. Depth profiles were obtained by sputtering with a Cs⁺ ion beam (3 keV). The analysis area was 100 × 100 μm². The mechanical property of the coating layer was investigated by atomic force microscopy (AFM, Bruker Corporation, Dimension Icon). A sample holder with an argon atomosphere was used to transport the electrode samples from the Ar-filled glovebox to the equipment used for the ex situ measurements.

## Electrochemical measurement

The fabrications of ASSLBs were carried out in the dry Ar-filled glovebox ($O_2 < 0.1$ ppm, $H_2O < 0.1$ ppm). Firstly, the cathode composites were prepared by a manual grinding process of different NMC811 cathode and LGPS powders in a weight ratio of 7:3 in an Agate mortar for 5 min. The solid-state electrolyte layer was prepared by pressing 70 mg of LGPS at 2 tons inside a polytetrafluoroethylene (PTFE) die (diameter of 10 mm) in a homemade KP-Solid cell. Then 10 mg of the cathode composites were dispersed on the surface of the solid-state electrolyte uniformly and pressed at 2 tons. The thickness of the cathode and solid electrolyte layers is around 40 μm and 400 μm, respectively. Finally, a piece of In foil (99.99%, ⌀ 10 mm, thickness 0.1 mm) was attached to the other side of the LGPS layer and pressed at

1 ton. The active NMC811 loading is about 8.92 mg cm$^{-2}$. There's no liquid electrolyte additive used during cell assembly. No external pressure is applied in the cell during electrochemical testing. The galvanostatic charge/discharge characteristics were conducted using a Land cycler (Wuhan, China) in a laboratory at 25 °C in the range of 2.7 V-4.3 V vs. Li/Li$^+$. The specific capacity refers to the mass of the active material in the positive electrode. For a single electrochemical experiment, two cells have been tested. Cyclic voltammograms (CV) were collected on a versatile multichannel potentiostation 3/Z (VMP3) using a scan rate of 0.05 mV s$^{-1}$ between 2.7 V-4.4 V vs. Li/Li$^+$. Electrochemical impedance spectroscopy (EIS) was also performed on the versatile multichannel potentiostat 3/Z (VMP3) by applying an AC voltage of 10 mV amplitude in the 7000 kHz to 100 mHz frequency range.

## Reporting summary

Further information on research design is available in the Nature Portfolio Reporting Summary linked to this article.

## Data availability

All data are available within the article and Supplementary Files, or available from the corresponding authors upon reasonable request. Source data are provided in this paper.

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

## Acknowledgements

This work was supported by the Natural Science and Engineering Research Council of Canada (NSERC), the Canada Research Chair Program (CRC), the Ontario Research Fund, the Canada Foundation for Innovation (CFI), Canada MITACS fellowships, the Canada Light Source at University of Saskatchewan (CLS), and the University of Western Ontario. M.G. wants to acknowledge the support from National Natural Science Foundation of China (M.G.), Shenzhen Science and Technology Program (Grant No. KQTD20190929173815000) (M.G.), Guangdong Innovative and Entrepreneurial Research Team Program (Grant No. 2019ZT08C044), National Natural Science Foundation of China (12004156) (Y.M.Z.), and the Shenzhen Basic Research Fund (JCYJ20190809181601639) (Y.M.Z.).

## Author contributions

The work was conceived and designed by J.L., X.L., and X.S. J.L. and X.L. fabricated the samples and tested the energy storage, stability, and other properties. The ALD coating was processed by S.D., Y.Z., Y.S, and R.L. The HEXPS was processed by Y.H., W.L., and T.S. The HRTEM and STEM images were processed by Y.Z., D.W., and M.G. The manuscript was drafted by J.L., X.L., Y.Z., and revised by J.L. and X.S. All authors participated in the data analysis and discussions.

## Competing interests

The authors declare no competing interests.
