## [Peer Review File · Nature Communications]

REVIEWER COMMENTS

Reviewer #1 (Remarks to the Author):

The paper describes a methodology to generate a gradient coating on NMC811 particles with the aim to produce a pathway to incorporate sulphur electrolytes with a stable interfaces that can be cycled without loss of capacity.

The authors have used a wide range of analytical techniques which proves that a kind of gradient coating is indeed achieved and that the coating somehow is also transferred to the grain boundaries of the NMC811 particles. Although the conditions of some of the techniques are not adequately described (i.e. some information about the conditions/equipment used for the ToF-SIMS should be provided). The set of results convincingly support their theory. There is however, a doubt about the homogeneity in their coating both in terms of chemical distribution and thickness which should be address with the measurement of at least a couple of regions of interest.

My main concern is related to the stability of the coating and its role preventing both mechanical and electrochemical degradation. The authors should provide further electrochemical testing including impedance analysis to prove interfacial resistance evolution and cycling at different voltages. The cycling temperature should also be provided for each of the measurements.

Have the authors analyse the effect of different thicknesses of the coating layer? if this is really an electronic insulator, what is the optimum thickness that can protect but not limit the cathode electrochemical activity?

Further analysis regarding mechanical stability and gas evolution would also be extremely helpful to evaluate the protective characteristic of the coating. It seems clear to me that there are some cracks developed in the cathode after cycling which would suggest that the coating won't prevent mechanical failure and this should be further discussed- what is the protective mechanism of the layer?

Reviewer #2 (Remarks to the Author):

This manuscript reports the surface modification of the cathode using sulfurized Li_3PO_4 that enables stable all-solid-state Li batteries. This work contains new ideas and valuable results that may interest readers studying all all-solid-state Li batteries. The authors tried to prove the structure of the coating layer formed on the surface through various detailed analyses. However, the author's explanation is a little confusing in several points. I think the following issues should be resolved before publication.

1. The authors claim that the gradual lithium oxy-thiophosphate ($\text{Li}_3\text{-}3\text{xPO}_4\cdot\text{xLi}_3\text{PS}_4$) interface is designed to ensure homogeneous Li^+ diffusion and completely eliminate the SCL layer due to its higher μ_{Li} near the region in contact with sulfide SSE. Could you prove that the Li mobility at the surface containing the $\text{Li}_3\text{-}3\text{xPO}_4\cdot\text{xLi}_3\text{PS}_4$ layer is higher than that of the surface containing the general oxide coating layer?

2. The authors refer to the sulfurized Li_3PO_4 coating layer as $\text{Li}_3\text{-}3\text{xPO}_4\text{-xLi}_3\text{PS}_4$ layer. It is clear that the Li_3PO_4 layer is gradiently sulfurized. However, isn't it difficult to be sure that the specific composition (Li_3PS_4) is formed?

3. The authors claim that the coating layer was formed at the grain boundary with the composition of $\text{Li}_3\text{-}3\text{xPO}_4\text{-xLi}_3\text{PS}_4$. Although it is true that S and P were observed at the grain boundary, unlike the surface, it is highly likely that a different composition was formed by reacting with Li or transition metal ions inside the cathode.

4. In the figures S7, S8, and S9, There is no figure caption explaining a, b, c, and d respectively. It is necessary to indicate those.

Response to the Comments of Reviewers' On "Gradient Interface on Surface and Grain Boundary of Ni-rich Layered Oxide Particles Enabling Highly Stable All-Solid-State Li Batteries" (Research article, *Nature Communications*, NCOMMS-22-14255A)

We highly appreciate the reviewers' recommendation on publishing this paper in *Nature Communications*. We are also very grateful for the reviewers to provide constructive comments and insightful suggestions for further improving the quality of the manuscript. The manuscript has been revised with great efforts. We carefully addressed the concerns and corrected any errors. The specific responses and revisions are listed below. Many thanks!

The most significant improvements in the revised version of the manuscript can be summarized as follow:

- We have evaluated the mechanical stability of the protective layer by atomic force microscopy (AFM) measurement;
- The impedance analysis of the In|LGPS|PS-LPO-NMC811 cell at different charge/discharge states to study the interfacial resistance evolution is shown;
- We have clarified the Li mobility at the coating layer;
- Other concerns have also been addressed.

REVIEWER REPORTS:

Referee: #1

Comments to the Author

The paper describes a methodology to generate a gradient coating on NMC811 particles with the aim to produce a pathway to incorporate sulphur electrolytes with a stable interfaces that can be cycled without loss of capacity. The authors have used a wide range of analytical techniques which proves that a kind of gradient coating is indeed achieved and that the coating somehow is also transferred to the grain boundaries of the NMC811 particles. Although the conditions of some of the techniques are not adequately described (i.e. some information about the conditions/equipment used for the TOF-SIMS should be provided). The set of results convincingly support their theory. There is however, a doubt about the homogeneity in their coating both in terms of chemical distribution and thickness which should be address with the measurement of at least a couple of regions of interest.

Response: We appreciate the reviewer's helpful suggestion.

We have added the detailed information about the condition/equipment used for the TOF-SIMS analysis in the revised manuscript and marked yellow.

“The TOF-SIMS measurements were conducted using a TOF-SIMS IV (ION-TOF GmbH, Germany) with a bismuth liquid metal ion source (25 keV). The base pressure in the analysis chamber is around 10^{-8} mbar. Depth profiles were obtained by sputtering with a Cs^+ ion beam (3 keV). The analysis area was $100 \times 100 \mu\text{m}^2$.”

Based on referee's suggestion, we further added the HAADF-STEM micrographs (as presented in Figure R1) of the PS-LPO-NMC811 sample in another region on the surface to

show the chemical distribution. We also added the HRTEM images (Figure R2-4) to show the thickness of the $\text{Li}_{3-3x}\text{PO}_4 \cdot x\text{Li}_3\text{PS}_4$ coating layer.

Figure R1. (a) The STEM-HAADF image of the PS-LPO-NMC811 particles section cut by focused ion beam, (b-h) the corresponding EDS elemental mapping, and (i) the overlay map.

Figure R2. HRTEM image of the PS-LPO-NMC811 sample.

Figure R3. HRTEM image of the PS-LPO-NMC811 sample.

Figure R4. HRTEM image of the PS-LPO-NMC811 sample.

1) My main concern is related to the stability of the coating and its role preventing both mechanical and electrochemical degradation. The authors should provide further electrochemical testing including impedance analysis to prove interfacial resistance evolution and cycling at different voltages. The cycling temperature should also be provided for each of the measurements.

Response: Many thanks for the suggestions. Based on reviewer's comment, we provide the operando impedance spectra of the In|LGPS|PS-LPO-NMC811 cell at different charge/discharge states in the first cycle. The obtained impedance spectra are shown in Figure R5. According to previous reports about the interfacial resistance for oxide cathodes in ASSLBs, the middle

semicircle in the impedance spectra should be assigned to the interfacial resistance between NMC811 and sulfide SSE ($R_{SSE/NMC}$)¹⁻². It can be seen from the spectra that the value of $R_{SSE/NMC}$ is only about 20 Ω after the first charge-discharge cycle, which is much smaller than that of 400 Ω for the Li-In| β -Li₃PS₄|NCM-811/ β -Li₃PS₄ cell reported.

To clarify this point, we also added this figure in the revised supporting information as Supplementary Figure 18 and marked yellow.

The cycling temperature was 25 °C for each of the measurements.

Figure R5. The operando impedance spectra of In|LGPS|PS-LPO-NMC811 cell cycled at 0.178 mA cm⁻² for 2 h and 2 h rest in the first cycle.

2) Have the authors analyse the effect of different thicknesses of the coating layer? if this is really an electronic insulator, what is the optimum thickness that can protect but not limit the cathode electrochemical activity?

Response: Many thanks for the suggestions.

The degree of sulfurization can be controlled by adjusting the mass ratio of P_4S_{16} to Li_3PO_4 . We have analyzed the effect of different thicknesses of the coating layer in our work. As mentioned in the manuscript, the degree of sulfurization can be controlled by adjusting the mass ratio of P_4S_{16} to Li_3PO_4 , thus the thickness of the sulfurized $Li_{3-3x}PO_4 \cdot xLi_3PS_4$ coating layer can be controlled. The PS-LPO-NMC811 sample obtained from 1 wt.% P_4S_{16} treatment showed a thin conformal coating with obvious P and S signals without altering the surface morphology of the NMC811 particles (Supplementary Figure 4), whereas a higher P_4S_{16} content of 5 wt.% led to thick and uneven surface film (Supplementary Figure 5). It's supposed that when the amounts of Li_3PO_4 and P_4S_{16} were unbalanced, the excess unreacted P_4S_{16} molecules would be deposited and accumulated on the surface of NMC811, which was unfavorable for the lithium ion migration. The PS-LPO-NMC811 cathodes with excess sulfurization (treated by 2.5 wt.% and 5 wt.% of P_4S_{16}) indeed showed inferior electrochemical performance (Figure R6, which also shown in Supplementary Figure 21). Thus, the optimum $Li_{3-3x}PO_4 \cdot xLi_3PS_4$ layer is around 10–20 nm in thickness with the LPO-NMC811 particles treated by 1 wt.% P_4S_{16} .

Figure R6. The (a) cycling performance and (b) corresponding Coulombic efficiencies of different PS-LPO-NMC811 cathodes prepared by different weight fractions of P_4S_{16} in the sulfurization process.

3) Further analysis regarding mechanical stability and gas evolution would also be extremely helpful to evaluate the protective characteristic of the coating. It seems clear to me that there are some cracks developed in the cathode after cycling which would suggest that the coating won't prevent mechanical failure and this should be further discussed- what is the protective mechanism of the layer?

Response: Many thanks for the suggestions. Based on reviewer's suggestion, we further evaluated the mechanical stability of the protective layer by atomic force microscopy (AFM)

measurement. The AFM image and corresponding DMT modulus image of the PS-LPO-NMC811 electrode are presented in Figure R7. The average DMT modulus of the $\text{Li}_{3-3x}\text{PO}_4\cdot x\text{Li}_3\text{PS}_4$ surface is about 2302 MPa. The low Young's modulus of such coating layer indicates its flexibility and softness to ensure precisely contact with the NMC811 particles to achieve conformal contact.³⁻⁴

Figure R7. (a) AFM image and (b) corresponding DMT modulus mapping of the PS-LPO-NMC811 sample.

We also added AFM analysis in the revised supporting information (Supplementary Figure 12) and the related description in the revised manuscript as highlighted.

“In addition, the mechanical analysis based on atomic force microscopy measurements (Supplementary Figure 12) proves the low Young's modulus of the $\text{Li}_3\text{P}_{1+x}\text{O}_4\text{S}_{4x}$ coating layer, which is propitious to achieve conformal contact with NMC811 particles.”

For the gas evolution, it is hard to test since the O_2 evolution of NMC811 cathode in liquid cell start at ~ 4.55 V (vs. Li^+/Li) (J. Phys. Chem. Lett. 2017, 8, 4820–4825), which is higher than the voltage we use here (4.4 V, vs. Li^+/Li). In liquid cell, the liquid electrolyte will promote the gas release based on the reaction between lattice O and organic solvent. While, in sulfide based

all-solid-state cells, the gas evolution is much difficulty. The active oxygen from lattice will react strongly with sulfide electrolyte instead of being released as oxygen. Even if it becomes oxygen, it still will oxidize the sulfide electrolyte and deposit in the interface.

As the reviewer pointed out, there are some cracks for the PS-LPO-NMC811 after 100 cycles as presented in Figure 7a. This indicates that the $\text{Li}_{3-3x}\text{PO}_4 \cdot x\text{Li}_3\text{PS}_4$ coating layer can't fully prevent the mechanical failure of the active NMC811 particles during long cycling. While combined multiple analyses and indeed highly improved electrochemical performance of the PS-LPO-NMC811 electrode in the all-solid-state battery systems, the fundamental protective mechanisms of the layer can be summarized as bellow.

First, the electrode/sulfide SSE side reaction is eradicated due to the conformal coating of the $\text{Li}_{3-3x}\text{PO}_4 \cdot x\text{Li}_3\text{PS}_4$ interfacial layer, which can avoid sulfide SSE decomposition and degradation. Thus, favorable Li^+ migration pathway can be ensured within the cathode composites. Moreover, the stable interfacial layer can also

Second, gradual Li^+ concentration and electrochemical potential can be guaranteed across the cathode/electrolyte interface, achieving good chemical compatibility of the coating layer with both the NMC811 cathode and sulfide SSE. The space charge layer caused by different chemical potential can be reduced, which will promote Li^+ migration across the interlayer

Third, the active cathode material inside the secondary particle is well preserved during the cycling of the PS-LPO-NMC811 electrode. In contrast, structural transformation from the layered to a spinel-like phase was found for the LPO-NMC811 electrode. The structural degradation phenomenon demonstrates bare LPO coating layer is not sufficient to protect the NMC811 particles for long-term cycling.

Referee: #2

Comments to the Author

This manuscript reports the surface modification of the cathode using sulfurized Li_3PO_4 that enables stable all-solid-state Li batteries. This work contains new ideas and valuable results that may interest readers studying all all-solid-state Li batteries. The authors tried to prove the structure of the coating layer formed on the surface through various detailed analyses. However, the author's explanation is a little confusing in several points. I think the following issues should be resolved before publication.

Response: Many thanks for your strong recommendation!

1) The authors claim that the gradual lithium oxy-thiophosphate ($\text{Li}_{3-3x}\text{PO}_4 \cdot x\text{Li}_3\text{PS}_4$) interface is designed to ensure homogeneous Li^+ diffusion and completely eliminate the SCL layer due to its higher μ_{Li} near the region in contact with sulfide SSE. Could you prove that the Li mobility at the surface containing the $\text{Li}_{3-3x}\text{PO}_4 \cdot x\text{Li}_3\text{PS}_4$ layer is higher than that of the surface containing the general oxide coating layer?

Response: Many thanks for the suggestions. The Li mobility at the $\text{Li}_{3-3x}\text{PO}_4 \cdot x\text{Li}_3\text{PS}_4$ layer is indeed difficult to measure since the layer is *in-situ* formed on the surface of the NMC811 particles and is only about 10-20 nm. However, the fast Li^+ mobility at this layer can be reflected and inferred from the following aspects:

1) The HAADF-STEM and HRTEM results proved that the $\text{Li}_{3-3x}\text{PO}_4 \cdot x\text{Li}_3\text{PS}_4$ layer is around 20 nm in thickness with some weakly crystalline clusters imbedded in the major amorphous phase. The clusters possess similar crystalline faces to that of the Li-argyrodite phase

(Supplementary Figure 11). Although the exact phase and composition of the clusters cannot be precisely determined, it proves that the reaction between Li_3PO_4 and P_4S_{16} can produce some substances with similar Li-argyrodite phase cluster. It has been reported that the Li-argyrodite phase is a highly Li^+ conductive phase, therefore, the $\text{Li}_{3-3x}\text{PO}_4 \cdot x\text{Li}_3\text{PS}_4$ coating should possess a higher Li^+ conductivity than the Li_3PO_4 layer and the bulk NMC811.

2) The GITT results show that the PS-LPO-NMC811 cathode presents the smallest polarization potential and the highest normalized $D_{\text{Li}^+} S^2 V_m^{-2}$ value during the entire discharge process, indicating its fastest Li^+ dynamics. As declared in the manuscript, the fast Li^+ migration for the PS-LPO-NMC811 cathode is mainly attributed to the significantly reduced SCL formation and the intrinsically high Li^+ conductivity of the $\text{Li}_{3-3x}\text{PO}_4 \cdot x\text{Li}_3\text{PS}_4$ coating layer. Indeed, the fastest Li^+ mobility is proved by the EIS as well as the GITT tests and further reflected from the best rate capability of the PS-LPO-NMC811 cathode (Figure 5g).

2) The authors refer to the sulfurized Li_3PO_4 coating layer as $\text{Li}_{3-3x}\text{PO}_4 \cdot x\text{Li}_3\text{PS}_4$ layer. It is clear that the Li_3PO_4 layer is gradiently sulfurized. However, isn't it difficult to be sure that the specific composition (Li_3PS_4) is formed?

Response: We appreciate the reviewer's helpful suggestion. Yes, it is indeed not easy to identify that the specific composition of Li_3PS_4 is formed. The formula of $\text{Li}_{3-3x}\text{PO}_4 \cdot x\text{Li}_3\text{PS}_4$ here is mainly to simplify the reaction between P_4S_{16} and Li_3PO_4 ($\frac{x}{4} \text{P}_4\text{S}_{16} + \text{Li}_3\text{PO}_4 \rightarrow \text{Li}_{3-3x}\text{PO}_4 \cdot x\text{Li}_3\text{PS}_4$, which can also be written as $\text{Li}_3\text{P}_{1+x}\text{O}_4\text{S}_{4x}$). Due to the electron-donating property of the sulfur-rich environment and S-S bridge bonds in the structure of the P_4S_{16} molecule, the sulfuration of Li_3PO_4 by P_4S_{16} can occur. We have shown in the manuscript that the $\text{Li}_{3-3x}\text{PO}_4 \cdot x\text{Li}_3\text{PS}_4$ layer is mainly amorphous, with some weakly crystalline clusters imbedded in the amorphous layer. The

clusters possess a similar crystalline face to the Li-argyrodite phase of Li_7PS_6 . It is difficult to quantify an exact composition due to the mainly amorphous nature and the gradient distribution of the Li, P, S, and O elements within this $\text{Li}_{3-3x}\text{PO}_4 \cdot x\text{Li}_3\text{PS}_4$ coating layer. However, the simplified formula here can keep the equation equilibrium of chemical reaction between P_4S_{16} and Li_3PO_4 and represents the total stoichiometric proportions of Li, P, S, and O within the layer.

To make the reaction and composition clearer and not cause misunderstanding, we changed the formula of “ $\text{Li}_{3-3x}\text{PO}_4 \cdot x\text{Li}_3\text{PS}_4$ ” to “ $\text{Li}_3\text{P}_{1+x}\text{O}_4\text{S}_{4x}$ ” in the revised manuscript and supporting information and marked yellow.

“It should be noted that the formula of $\text{Li}_3\text{P}_{1+x}\text{O}_4\text{S}_{4x}$ here is mainly to simplify the reaction between P_4S_{16} and Li_3PO_4 ($\frac{x}{4}\text{P}_4\text{S}_{16} + \text{Li}_3\text{PO}_4 \rightarrow \text{Li}_3\text{P}_{1+x}\text{O}_4\text{S}_{4x}$) while not the specific composition.”

3) The authors claim that the coating layer was formed at the grain boundary with the composition of $\text{Li}_{3-3x}\text{PO}_4 \cdot x\text{Li}_3\text{PS}_4$. Although it is true that S and P were observed at the grain boundary, unlike the surface, it is highly likely that a different composition was formed by reacting with Li or transition metal ions inside the cathode.

Response: Many thanks for the suggestions. We agree with the reviewer's opinion that a different composition might be formed at the grain boundary by reacting with Li or transition metal ions inside the cathode. However, it is quite difficult to characterize and determine the exact composition at the grain boundary. The most popular method is using HADDF-STEM and HRTEM measurement after focused ion beam cutting of the particles as presented in the manuscript. While it should be noted the characterization methods are quite limited to detection area and cannot give the whole information of the grain boundary.

Based on referee's comments, we have added the description about the difference between surface and grain boundary in the revised manuscript and marked yellow.

“The compositions at the grain boundary might be slightly different from that of the surface due to the possible diffusion of the transition metal ions inside the NMC811 particles.”

4) In the figures S7, S8, and S9, There is no figure caption explaining a, b, c, and d respectively. It is necessary to indicate those.

Response: Many thanks for the suggestions. We have added the figure caption explaining of the Supplementary Figure 7-9 in the revised supporting information and marked yellow.

“**Supplementary Figure 7.** The (a,b) cross-sectional HRSTEM and (c-f) HRTEM images of the LPO-NMC811 sample, the clear cross-section was cut by focused ion beam after ALD coating process.”

“**Supplementary Figure 8.** (a) The HAADF-STEM image of the LPO-NMC811 particles, (b-g) the corresponding EDX elemental mapping of Ni, Co, Mn, O, P, C, and (h) the overlay map of these elements.”

“**Supplementary Figure 9.** (a) The STEM-HAADF image of the PS-LPO-NMC811 particles section cut by focused ion beam, (b-h) the corresponding EDX elemental maps of Ni, Mn, Co, O, P, S, C, and (i) their overlay map.”

Thank you again!

Reference

1. Koerver, R.; Aygün, I.; Leichtweiß, T.; Dietrich, C.; Zhang, W.; Binder, J. O.; Hartmann, P.; Zeier, W. G.; Janek, J. r., Capacity fade in solid-state batteries: interphase formation and chemomechanical processes in nickel-rich layered oxide cathodes and lithium thiophosphate solid electrolytes. *Chemistry of Materials* **2017**, *29* (13), 5574-5582.
2. Kim, A.-Y.; Strauss, F.; Bartsch, T.; Teo, J. H.; Hatsukade, T.; Mazilkin, A.; Janek, J. r.; Hartmann, P.; Brezesinski, T., Stabilizing effect of a hybrid surface coating on a Ni-rich NCM cathode material in all-solid-state batteries. *Chemistry of Materials* **2019**, *31* (23), 9664-9672.
3. McGrogan, F. P.; Swamy, T.; Bishop, S. R.; Eggleton, E.; Porz, L.; Chen, X.; Chiang, Y. M.; Van Vliet, K. J., Compliant Yet Brittle Mechanical Behavior of Li₂S–P₂S₅ Lithium - Ion - Conducting Solid Electrolyte. *Advanced Energy Materials* **2017**, *7* (12), 1602011.
4. Zhou, X.; Zhang, Y.; Shen, M.; Fang, Z.; Kong, T.; Feng, W.; Xie, Y.; Wang, F.; Hu, B.; Wang, Y., A Highly Stable Li-Organic All-Solid-State Battery Based on Sulfide Electrolytes. *Advanced Energy Materials* **2022**, *12* (14), 2103932.

REVIEWERS' COMMENTS

Reviewer #1 (Remarks to the Author):

I believe that the authors have adequately responded to my concerns and therefore I recommend publication of the paper in its actual form

Reviewer #2 (Remarks to the Author):

Authors have made changes to address previous concerns, and I would recommend the publication of this manuscript.